# Pedestrianization Impacts on Air Quality Perceptions and Environment Satisfaction: The Case of Regenerated Streets in Downtown Seoul

**DOI:** 10.3390/ijerph181910225

**Published:** 2021-09-28

**Authors:** UnHyo Kim, Jeongwoo Lee, Sylvia Y. He

**Affiliations:** 1Department of Urban Design and Studies, Chung-Ang University, Seoul 06974, Korea; freddie3944@gmail.com; 2Department of Geography and Resource Management, The Chinese University of Hong Kong, Shatin, N.T., Hong Kong, China; sylviahe@cuhk.edu.hk

**Keywords:** pedestrianization, walking environment, air-quality perception, satisfaction, greenway

## Abstract

Previous studies have investigated the increased volume of pedestrians to establish success rates of the pedestrian-friendly policy after a street redesign intervention. However, few studies have focused on the effect of street regeneration on air quality perception and user satisfaction. The influence of the physical environment on street vitality may vary, depending on area context and regional factors. A comprehensive understanding of effective interventions could increase pedestrians’ satisfaction with their walking environment. This study examines the effect of pedestrianization on individuals’ air quality perception and satisfaction, based on three regenerated streets in Seoul, Korea. We analyzed data from 672 questionnaires administered after the pedestrianization project. We used a subset of variables in a binary logistic regression model to understand general determinants of user satisfaction toward their walking environment. Our case study contributes to the verification of pedestrianization effects on air quality perceptions. Results show that overall satisfaction could be acquired through positive perceptions of air quality, as achieved through pedestrianization of streets. Moreover, pedestrian satisfaction varies according to the purpose, activities and health-related behaviors and attitudes. The interrelationships between environmental health, activity, satisfaction and quality of life provide design insights to consider when implementing pedestrianization projects in the future.

## 1. Introduction

### 1.1. Pedestrianization

Cities have changed their planning strategies to reorganize the urban structure from car-dependent to pedestrian-oriented designs over the past few decades [1,2,3]. As a result, pedestrian-friendly environments predominantly limit the number of cars and widen the sidewalks in favor of pedestrian movement [4]. This is effective for promoting physical and social activities [5,6,7,8], as well as mitigating both air and noise pollution [9,10,11].

Pedestrianization is defined as converting an existing road into an area for pedestrians alone [6]. Pedestrianization has been carried out in many countries to create a walkable environment. Developed countries in the western world started creating pedestrian spaces to regenerate and prevent further decline of downtown commercial areas from the 1950s to 1960s. It was used as a method of enhancing livability and promoting pedestrian activities through pedestrian malls located in the center of cities in Germany, Denmark and, later, North America [7,12,13]. Pedestrianization has also appeared in Asian cities with a slightly different context. Due to a shortage of urban space within the city center because of rapid urbanization and the motorization process, underground and elevated walkways were proposed to accommodate pedestrian mobility in the city’s dense urban core [14,15]. Furthermore, as more private vehicles resulted in severe air pollution, cities began to change strategies, from expanding road capacities to prioritizing the pedestrians and creating walkable environments [16].

In South Korea, starting with Seoul from the late 1990s, the creation of a pedestrian-friendly environment is constantly being emphasized as one of the major policy measures of local governments [17]. These pedestrianization projects have been conducted steadily as a regeneration tool on a street level in many old towns [18]. The importance of pedestrianization for urban regeneration has been mainly proven by outcomes such as revitalization of the local economy, increases in the number of start-ups and pedestrian volume [19]. However, street transformations have often had undesirable consequences in terms of gentrification and social exclusion [20,21,22]. In addition, it has been pointed out that some regeneration projects focusing on physical refurbishment of urban streets often neglect the culture of the place and sensibility of the residents [23]. Despite this dark side of regeneration, pedestrianization is still an efficient measure for the urban core in terms of cost and time invested [10,24], as compared to regional-level characteristics that cannot easily induce change in existing cities. In addition, pedestrianization is an effective measure to revitalize recreational spaces. This is because, unlike walking for commuting, leisure walks are influenced by the environment at the street level, which is the area of direct perception [25,26].

### 1.2. The Walking Environment and Perceptions

Most existing studies focused on the physical environment when examining the factors affecting citizens’ physical activity. They demonstrated the impact of the mesoscale-built environment by focusing on the “3Ds”-centered (density, diversity and design) [27] physical environment for factors at a neighborhood level [26,28,29,30,31]. Further, traffic volume and street networks have also been effective factors influencing physical activity at a neighborhood level [32,33,34]. These are the environmental characteristics formed at the regional or neighborhood levels that increase active travel by increasing the probability of residents choosing to walk and bike—instead of drive—in terms of effectiveness.

Some studies investigate potential factors that may affect pedestrian perception at the micro-scale (or street) level. These factors include street infrastructure, sidewalk width and pavement, distance-to-height (D/H) ratio of the street and presence of street trees [35,36,37,38]. These are the environmental characteristics formed at street units. Pedestrians experience these factors directly and instantaneously through their senses, while using the street space and engaging their environmental perceptions. These characteristics directly affect feelings and preferences. Thereby, they determine their intentions of using the street and their walking activities.

In recent years, studies have expanded to include perceptions, which play a mediating role between the physical environment and walking behaviors [35,39,40,41,42,43,44]. Numerous studies examined the role of individual environmental features in impeding or supporting walking behavior. Moreover, the latest literature focuses on the mediating role of internal states, noting that residents’ perceptions of the environment are not an assemblage of individual images of spatial elements, but an integrated sense of the street, containing elements such as comfort, convenience and diversity [40,42,44,45]. Further, individuals perceive the same environment differently, depending on the factors they perceive as important. Safety was also derived as a significant factor toward the intention to walk [46,47,48]. Studies conducted on streets in downtown Seoul, Delhi and other metropolitan cities with dense environments pointed out that comfort and vitality of the street are key factors determining pedestrian behavior [49,50]. Such studies have shown that pedestrians perceive the traffic condition of the street as a key factor. This includes vehicle safety and the convenience of walking or using public transportation. However, walking behavior and pedestrian intentions are the results of a diverse and multifaceted perspective. Given the situation of metropolitan cities characterized by traffic and high-rise building densities, most citizens are concerned about exposure to congestion and air pollution. In the large cities of Asia, the perceived threat of air pollution and the possibility of adverse health effects are increasing [51,52].

### 1.3. Air Quality Perceptions

Studies have shown that air quality is perceived differently according to individual characteristics. Age and health status were found to influence individuals’ perception of air quality [53,54,55]. Edgley et al. [53] found that older groups generally perceive air quality negatively. Smoking history also correlates to air quality perception [56]. Furthermore, people with asthma perceived air quality more negatively than those without asthma. Self-reported health status and the perceived physical environment were also found to influence air quality perception. According to a study by Simone [57], people who considered themselves to be healthy were more likely to perceive air quality positively and vice versa. The social environment also affects the perception of air quality. For example, if people perceive their neighborhood as poor, they tend to perceive the air quality negatively [58]. These results indicate that air quality perception may vary among individuals because of their individualistic views of the phenomenon [59].

Past research enriched our understanding of how the built environment influences perception and activities. However, certain research gaps still exist. Few studies analyze the effects of pedestrianization on perception and satisfaction levels. Although previous studies have investigated the increase in the volume of pedestrians after area interventions to assess the degree of success of the pedestrian-friendly policies, few studies have focused on the effect of pedestrianization on air quality perception and satisfaction. In addition, most research investigating the impact of the implementation did not rule out that the influence of the physical environment on street vitality may vary, depending on the context of the area and regional factors. These research gaps may hinder our comprehensive understanding of the effective intervention for increasing intention and satisfaction toward their walking environment. Thus, this study aims to address these gaps by examining the effect of pedestrianization on individual air quality perception and satisfaction by focusing on regenerated streets in Seoul, South Korea.

If we compare the average air quality indices of the 13 largest metropolitan cities of 2016, Seoul had the highest level of air pollution [60]. Thus, concerns about the resultant health threats were high. According to a survey on the perception of air quality, approximately 84% of Seoul citizens report daily discomfort due to the fine dust in the city and around half of the citizens said that they refrained from going out [61]. This indicates that air pollution has become a significant factor threatening Seoul’s citizens’ daily life and outdoor activities.

A variety of policy measures aimed at reducing air pollution are currently being implemented in Seoul. Seoul has continued its efforts to become a pedestrian-friendly city by reducing traffic and reorganizing streets to prioritize pedestrians and provide green spaces [17,62]. Most of Seoul’s pedestrianization policy measures were conducted as a street-redesign intervention. These included pedestrian-friendly streets and the pedestrian-priority zones. In addition, many street-environment maintenance projects are being carried out under this initiative. One study examined changes in the number of visitors, vacant stores and sales after completing such projects [63]. Although this research used data that can objectively measure the level of street vitality, pedestrian satisfaction is key when evaluating the effect of implementation from an individual’s perspective. Furthermore, according to the results of a study analyzing the effects of street-redesign intervention, measurements were inconsistent, even though most projects focused on reducing car lanes, expanding pedestrian areas and improving pavement infrastructure. This implies that the physical characteristics that create the street environment can have different effects, depending on the conditions and context of the street or the area.

This study aimed to examine whether implementation strategies such as pedestrianization could be substantially linked with stimulating walking and creating pedestrian satisfaction. Recreational walking is more sensitive to satisfactory pedestrian experiences, whereas utilitarian walking is less responsive to environmental quality [41,43,64]. Therefore, sites were selected, taking into consideration areas where recreational walking trips are prevalent and representative streets are regenerated in the middle of Seoul downtown. These sites are (1) Sejongno, which was recreated as a pedestrian plaza by reducing 16 traffic lanes to 10 and transforming it into a public space; (2) Cheonggyecheon-ro, which was rebuilt as a linear park by demolishing an overpass and restoring the stream; (3) Seoullo, 7017 Skygarden, which was converted from the Seoul Station Overpass into a pedestrian path. At all three sites, there is a trend of pedestrianized greenway footpaths located in the middle of the street, parallel to the sidewalk path.

The following hypotheses were established: (1) Pedestrians perceive the air quality to be better when walking on a greenway footpath, rather than on the sidewalk, even though the footpath and sidewalk are parallel and located on the same street (this was proposed to control the effect of the different street environment); (2) Satisfaction and intention to walk are based on a set of factors relating to pedestrians’ characteristics, health-related behaviors, and attitudes and perceptions regarding their own built environment and air quality.

## 2. Materials and Methods

### 2.1. Description of the Study Area

Three pedestrianization projects were selected to examine the pedestrian perceptions of different sites: Sejongno, Cheonggyecheon-ro and Seoullo (Table 1). The sites are located in the center of downtown Seoul and have a common characteristic, in that the pedestrian roads on both sides are parallel to the pedestrianized greenway footpaths in the middle (Figure 1). To compare the different perceptions by different pedestrian path designs, surveys of the greenway footpath were compared with simultaneous surveys conducted with random participants of the sidewalks.

Sejongno is one of the symbolic spaces of Seoul. Major government offices are located here and it carries heavy traffic. Sejongno greenway was created in the middle of the Sejongno roadway by removing six-lane roads. Its footpath is on the same level as the sidewalk and is efficiently connected to the sidewalk by crosswalks. Cheonggyecheon-ro was converted into an urban stream after demolishing a four-lane urban highway. It has been praised as a successful example of achieving urban regeneration and ecological sustainability. The area is often used as a leisure and rest area for citizens and it offers a variety of activities. Below sidewalk level, there is a pedestrianized greenway footpath along the reconstructed urban stream that can be reached by the stairs. Seoullo is an area where an old overpass near Seoul Station was transformed into a pedestrian park. The Seoul Station previously formed the center of heavy traffic downtown and there was insufficient space for pedestrians. Nowadays, it has become a popular open space for both residents and tourists. The pedestrianized greenway footpath of Seoullo is located on an overpass about 10 m above the sidewalk and can be reached by elevator or escalator.

The visitors’ main purpose for visiting the three abovementioned regenerated streets differed (Figure 2). Many people met to conduct business or social activities at Sejongno, where public events and gatherings frequently occur. This was more evident for the sidewalk adjacent to the building than for the greenway footpath. In Cheonggyecheon-ro, most visits were related to leisure and social activities, for both the greenway footpath and sidewalk areas. The pedestrianized greenway footpath of Cheonggyecheon-ro was frequented for physical activities, such as walking and exercising, while the sidewalks were most frequently used for commuting or transferring. Regarding the greenway footpath in Seoullo, about half of the pedestrians visited the street to walk or exercise. This indicates that people who walked on the greenway footpath were often recreational walkers, while those who walked on the sidewalk path often did so for utilitarian purposes.

All in all, three sites have a similar pattern of visit frequency by the purpose. A total of 35–45% of greenway footpath users visit the sites for the purpose of physical activity, including walking. Approximately 55% of sidewalk users visit the sites for the purpose of social or business commute trips. All these locations have the lowest frequency of trips for shopping or transit purposes. The homogeneity of the study sample (i.e., distribution by trip purpose) helps to generalize the study results.

### 2.2. Data

A field survey was conducted from 6 July to 24 July 2019. As individuals’ perception of the characteristics of the streets may vary between weekdays and weekends, surveyors collected responses once on a weekday and once on a weekend for each site. Pedestrians were randomly selected for the survey.

The survey questionnaire consisted of three parts: demographic characteristics; propensity and activity level; perception of the street environment. Personal characteristics included gender, age and occupation. Individual propensity and activity characteristics included the frequency of street visits, the purpose of the trip, activity level and health-related behavior and attitudes. The third part was related to perceptions regarding the street environment where pedestrians walked.

A total of 672 questionnaires were completed and 621 valid questionnaires were analyzed after excluding 51 questionnaires with missing values. The number of female respondents (56.3%) was slightly higher than that of male respondents (43.8%). The number of questionnaires collected from each target site was 210 for Sejongno (33.8%), 220 for Cheonggyecheon-ro (35.4%) and 191 for Seoullo (30.8%). Of the sample, the greenway footpath (of all three sites) accounted for 45.9% of responses and the sidewalk accounted for 54.1%, which is an approximately even distribution. Notably, there is a variation between the characteristics of people using greenway footpaths and sidewalk paths. The daily visitors who frequent the site were around 12% higher for the sidewalk than for the greenway footpath. One-fifth of the respondents used the sidewalk for commuting, while only 8.4% of pedestrians used the greenway footpath for that purpose (Table 2).

This study assumed that personal and environment characteristics influence pedestrians’ satisfaction with their walking environment. Pedestrian satisfaction of the street environment was assessed using a six-point Likert scale with items ranging from 1 = not at all dissatisfied to 6 = very satisfied. These categories were condensed to create a dichotomous variable (not satisfied/satisfied) for a binary logistic regression analysis, taking advantage of the model that allows for exploratory power in predicting the odds of the dependent variable.

The variables used in our binary logistic regression were classified into three groups: locational, individual and perceived environment characteristics. The site location and greenway footpath factors were added as dummy variables to compare the differences in physical environments. As perception may vary depending on the air quality and date of the survey, the measured air quality data from official monitoring stations and whether the survey was conducted on a weekday or weekend were added as control variables.

As an individual characteristic, activity level was assessed using walking time, with respondents reporting their average walking travel time on a weekday. The health status of an individual was measured through respiratory disease symptoms, such as asthma and habitual smoking. Other health-related attitudes were measured using a six-point Likert scale using sensitivity to the air quality and individuals’ attitudes toward traffic control measures for better environmental health (Table 3).

This study builds on previous studies [39,42,45,46,47,65] and constructed variables for the perceived street environment; these included perceived air quality and noise levels, which are known to influence the quality of life. Twenty-four other variables of perceived environment were entered into the survey and these variables were scored on a four-point Likert scale. A principal component analysis (PCA) was used to reduce the multiple overlapping perceptional variables to five underlying factors: vitality, comfort, restorativeness, connectivity and lack of congestion.

## 3. Results and Discussion

### 3.1. Differences in Air Quality Perceptions

The study examines how air quality perceptions vary according to the pedestrian path design. First, a Mann–Whitney analysis was performed to investigate the differences in air quality perception between the greenway footpath and sidewalk (Table 4). The result shows that pedestrians perceive their walking environment differently depending on the structure of the street. This result was significant in Cheonggyecheon-ro and Seoullo and not significant in Sejongno. This could be because of the different structures of the greenway footpath. The formation of the pedestrianized greenway footpaths in these three sites was similar, as all three are located in the middle of the street. However, there is a key difference, consisting in Cheonggyecheon-ro and Seoullo being vertically separated from road traffic; thus, pedestrians cannot see vehicles. However, Sejongno has road traffic on the same level. The results are consistent with previous findings [66,67] that state that perception differs according to the visibility of pollutants, such as vehicle traffic.

### 3.2. Comparison of Pedestrian Perceptions by Pedestrian Path Design

#### 3.2.1. Components of the Perceived Street Environment

This study investigates the effectiveness of each site’s pedestrianization and its effect on street-environment perception, based on its ability to trigger the intention of walking. To reduce the multiple overlapping perceptional variables into components that characterize the perceived street environment, a PCA was conducted with varimax rotation. Among the 24 perceptional variables that explain the street environment, three variables (with a commonality of 0.5 or less) were excluded. Finally, a PCA was performed on 21 variables (see Table 5). Items were reduced to a five-factor solution with a total variance of 62.02%. The minimum eigenvalue for the selected components was 1.0, where the Kaiser–Meyer–Olkin value was 0.886, indicating adequate sampling [47]. Cronbach’s Alpha was also high, ranging from 0.705 to 0.835. The results are shown in Table 5.

A factor score was calculated as the weighted sum of an individual’s scores on 21 perceptional variables, explaining 62% cumulative variance of the total parameters. Factor 1 (vitality) accounts for 15.6% of the variance. It comprises five variables: lively atmosphere, many cultural elements and attractions, a symbolic place, interesting and harmonious buildings, and attractive landscape. Factor 2 (comfort) accounts for 15.2% of the variance. It consists of the following items: calm and quiet, clean, safe from traffic, convenient and easy to walk on, comfort, sufficient rest spots and well landscaped. Factor 3 (restorativeness) accounts for 12.1% of the variance. It consists of items related to the relaxation space and mental healing functions, as follows: sufficient trees and shade, sufficient seating, sufficient green space, not monotonous/boring. Factor 4 (connectivity) accounts for 9.7% of the variance. It comprises wide sidewalks, streets that physically connect and no parked cars and bicycles on the streets. Finally, Factor 5 (lack of congestion) accounts for 9.5% of the variance. It consists of the following items: not noisy and not congested with traffic. The Cronbach alpha value for each factor was above 0.7, indicating the high level of internal reliability.

#### 3.2.2. Differences in Street Perceptions

We compared the differences in perception of street environments (by greenway footpath and sidewalk) by examining the five factors derived previously (Figure 3). In Sejongno, the greenway footpath seems to influence pedestrians to recognize street environments as uncongested, compared to sidewalks. However, the difference between sidewalks and greenway footpaths in vitality, comfort, restorativeness and connectivity was small. This is because the Sejongno pedestrian square (greenway footpath) is built on the same level as traffic in the center of the busy lane; therefore, it is continuously exposed to vehicle noise and exhaust fumes. In addition, there is no resting space for pedestrians in the form of benches and street trees.

Cheonggyecheon-ro contrasts with Sejongno in that pedestrians perceived the greenway footpath more positively than the sidewalks in terms of vitality, comfort, restorativeness and connectivity. This is because there is an urban stream alongside the greenway footpath, where various events—such as the lantern festivals and busking by the greenway footpath—take place, which is in harmony with stream. Therefore, pedestrians perceive the greenway footpath as enjoyable and lively, compared to the sidewalk. Moreover, there are several green rest areas by the stream, which allows pedestrians to feel relaxed and comfortable.

Among the three sites, Seoullo showed the most significant difference between the sidewalk and the greenway footpath. Differences were particularly evident in the aspects of vitality, comfort and being uncongested. Possible reasons for the difference between the sidewalk and greenway footpath are as follows. First, the greenway footpath of Seoullo has a higher comfort level than the sidewalk. The greenway footpath of Seoullo is elevated approximately 10–14 m above ground level and its pedestrians are not directly exposed to noise and exhaust fumes from vehicles. This allows the pedestrian to perceive the greenway footpath as more comfortable and pleasant. Second, the greenway footpath of Seoullo shows a higher restorativeness level than the sidewalk. There are no rest spaces along the sidewalk and the scenery from the street is also monotonous. For the Seoullo greenway footpath, there is a rest area at every 300 m and benches and vegetation are abundant. Third, the greenway footpath of Seoullo also shows a higher vitality level than the sidewalk; this is because there are many activities to enjoy along the greenway footpath, such as flea markets and busking. Conversely, the sidewalk is narrow and there are no rest spaces on the street. Thus, pedestrians find it monotonous and stuffy. Furthermore, the greenway footpath appears to have lower connectivity than the sidewalk. According to Sim et al. [68], connectivity is an important characteristic of an elevated park because it allows pedestrians to access the surrounding areas [69]. In this respect, not only is the greenway footpath of Seoullo difficult to access from the sidewalk, it is also not connected to the surrounding areas. Therefore, connectivity is lower than sidewalks, where connection points appear more frequently during the walking experience.

Overall, the pedestrianized greenway footpath has improved the positive perception of the street environment, especially in terms of comfort, vitality and an uncongested space. However, it varies among different greenway designs. Pedestrians in Sejongno did not perceive the greenway footpath as a better space than the sidewalk, whereas, in Cheonggyecheon-ro and Seoullo, it is apparent that pedestrians perceive the greenway footpath more positively. In the case of Seoullo with its heavy traffic volume, the vertical separation created by the elevated greenway footpath was effective in increasing satisfaction levels with the street environment. In the case of Cheonggyecheon-ro, alongside the Cheonggye stream, perceptions of the sidewalk path and the greenway footpath were both relatively positive, because the greenway footpath is near the urban stream and in sight of pedestrians walking on the sidewalk as well. This result demonstrates that the satisfactory level of pedestrians vary, depending on the pedestrian path design for specific context of location.

### 3.3. Factors Affecting Pedestrian Satisfaction

Table 6 shows the results of the binary logit regression analysis by exploring factors related to street satisfaction. Model 1 was constructed to include the physical environment factors of the greenway footpath, location factors, situational variables (e.g., weekday) and the air quality value from two official monitoring stations at nearby research sites. In Model 2, we included pedestrian characteristics, trip purpose, walking activities and individual health-related behaviors and attitudes. To explore the relationship between perceived aspects and street satisfaction, we added the perceived street environment factors in Model 3. For the perceived street environment factors, we included five factors derived earlier by PCA. All of the variables, including perceived air quality and noise level factors, were assessed in Model 4.

To examine locational factors in Model 1, the greenway footpath positively affected pedestrians’ street satisfaction levels. Pedestrians using the greenway footpath perceived the street environment more positively than pedestrians using the sidewalks. Furthermore, the model indicated that location factors are also associated with street satisfaction. Pedestrians in Cheonggyecheon-ro were more likely to perceive the street environment positively than pedestrians in Sejongno, which is the reference variable. There was no significant difference in Seoullo when compared to Sejongno.

Individual pedestrian factors were examined in Model 2. The results showed that people in their 20s were more likely to experience street satisfaction, whereas there was no significant difference when comparing people in their 60s with those in their 30s–50s. This finding aligns with those of Sahani [70], who found that younger pedestrians tend to be more satisfied with the street environment than middle-aged or older people. Similarly, Lee [71] also found that the street satisfaction of people in their 20s is higher than for those in their 40s in the pedestrian specialization of streets. Our results also show that people who visited the street for leisure activities were more likely to experience higher satisfaction levels with the street environment. This result is consistent with the studies that argued that utilitarian and recreational purposes should be distinguished [32,72,73].

We add evidence to the existing literature, which emphasizes the importance of controlling for individual characteristics related to the propensity to walk and usual physical activity amount [74,75]. The amount of time he or she spent walking was found to exert a negative effect on satisfaction levels. People who spent less time walking were more likely to evaluate their street satisfaction positively. However, no consistent conclusion has been drawn regarding the correlation between average walking time and street satisfaction. St-Louis et al. [76] stated that active people tend to evaluate their street environment positively, whereas Kari [77] stated that there is no correlation between the evaluation of the street environment and usual walking habits. As these studies show contradictory results, further research is necessary to establish a correlation between walking time and street satisfaction.

This study investigates whether individual health-related variables, including respiratory disease (e.g., asthma) and smoking habits, significantly affect environment satisfaction [56]. The result shows that people who smoke perceive street satisfaction more negatively than those who do not. Optimistic attitudes toward traffic control measures had the most significant positive effect on pedestrians’ street satisfaction. Those who supported traffic control measures were 3.8 times more likely to be satisfied with the street environment than those who opposed the policy. This result is in line with research that suggests that environmental awareness significantly influences physical activity, indicating that pedestrian attitudes play an important role in walking behavior [78,79].

Models 3 and 4 explored the relationship between perceived aspects of pedestrians and street satisfaction levels. The chi-squared value increased significantly—from 88 (Model 3) to 171 (Model 4)—with the perceived street environment factors. This indicates that the influence of the subjective environment factors, which belongs to the perceived aspect of the pedestrians, displays a significant impact on street satisfaction. Among the five factors of perceived street environment, all but the connectivity factor impacted street satisfaction. The other four factors (i.e., vitality, comfort, restorativeness and lack of congestion) have different effects on street satisfaction. The vitality and comfort levels of the street were associated with a significantly higher probability of street satisfaction, with an odds ratio close to 2, compared to restorativeness (0.7) and lack of congestion (0.7) levels (Model 3; Table 6). Connectivity of the street-on-street satisfaction was not significant in this model. Even though connectivity has been recognized as an important factor for improving mobility near subways and in areas with traffic congestion [46,47,48], it does not appear to be a significant factor in the perception of pedestrians in this study, as the ratio of visitors for leisure travel purposes (e.g., shopping, appointments) accounted for around 40%.

Model 4 enhances Model 3 by including variables related to perceived air quality (air quality and noise perceptions), increasing the explanatory power. This result illustrates that, despite the lack of data regarding perceptions of air quality, it is an important factor when evaluating the street satisfaction of pedestrians. Furthermore, both air quality and noise perceptions were found to influence street satisfaction levels significantly. Pedestrians who perceive air quality positively were 2.5 times more likely to perceive overall street satisfaction positively when all other variables were controlled for. This result suggests that the perceived air quality of the street can play an important role in improving pedestrian street satisfaction. In addition, as air quality perception in the street is closely related to traffic volume, we included noise perception in our study and found that the perceived level of noise negatively influenced street satisfaction.

## 4. Conclusions

In recent years, Seoul has implemented various pedestrianization projects that reorganize roads into greenway footpaths, providing space for pedestrians and sufficient green spaces in the city. We conducted this empirical study focusing on the perception of pedestrians in the three pedestrianization areas where roads were reorganized into greenway footpaths. The sites were Sejongno, where the greenway footpath was built on the same level as the existing sidewalk; Cheonggyecheon-ro, where the greenway footpath was built on a level below the road along the Cheonggye stream; Seoullo, where the greenway footpath was built on the elevated highway in a heavy-traffic area. The three sites are characterized by different forms of greenway footpaths that have been regenerated through the pedestrianization project. The implications of this study are as follows.

First, this study identified the effect of pedestrianization on air quality perceptions and user satisfaction. In our study, the sidewalks and greenway footpaths were parallel, sharing the same environment in terms of land use and traffic volume; thus, the pure pedestrianization effect on user perception could be compared. This precise measurement was lacking in previous studies and can help determine future policy and planning that correlates with pedestrian satisfaction. By comparing three different forms of pedestrianization methods, we also found that the location characteristics and the formation of the greenway footpath play a significant role in user perception.

Second, the perceptional aspect of pedestrians was noted as a significant factor in mediating the relationship between the street environment and user satisfaction. Numerous studies addressed the relationship between the physical environment of streets and walking behavior; however, various perceptional aspects that directly affect walking intention were overlooked. We found that users of pedestrianized greenways were comparatively most satisfied with vitality, comfort and restorativeness, while least satisfied with connectivity. From an urban design perspective, providing more pedestrianized greenway paths in downtown Seoul is essential to promote daily walking and physical activity, given that citizens feel that pedestrianization streets are environmentally healthier and more suitable for recreational activities.

Third, this study found that overall satisfaction could be acquired through positive perceptions of air quality, which can be achieved through the pedestrianization of streets. In fact, given that fine dust and air pollution are mentioned as the number one factor hindering walking behavior [80], both actual and perceived levels of air quality improvement are important conditions of street environment. Furthermore, the pedestrian satisfaction degree varies according to health-related behaviors and attitudes. The interrelationships among environmental health, activity, satisfaction and the quality of life provide design insights to consider when implementing pedestrianization projects in the future.

There are limitations to this study that future studies should supplement. One of the limitations of our study relied on cross-sectional data obtained from the survey conducted during the day, in summer 2019. This measurement does not fully account for those people taking a trip during the winter, nor does it account for those people walking at night in the pedestrianization street. The variations by season and year and time of day must be carefully considered in the future analysis. In addition, although this study was conducted prior to the COVID-19 pandemic, further research on the healthy public spaces of the post-pandemic city is intriguing. It would be interesting to explore whether the pedestrianization of streets results in a feasible solution to improve mental and physical health with low pollution during the pandemic [81], given that the existing research supports the assertion that the pedestrianization project reduces the exposure of pedestrians to air pollution [82]. Finally, this empirical study used three sites of the downtown area of Seoul as the study area. Therefore, it is difficult to generalize the results of this study to areas other than downtown Seoul. Nevertheless, these findings have implications for street redesign intervention for other cities with similar characteristics.

## Figures and Tables

**Figure 1 ijerph-18-10225-f001:**
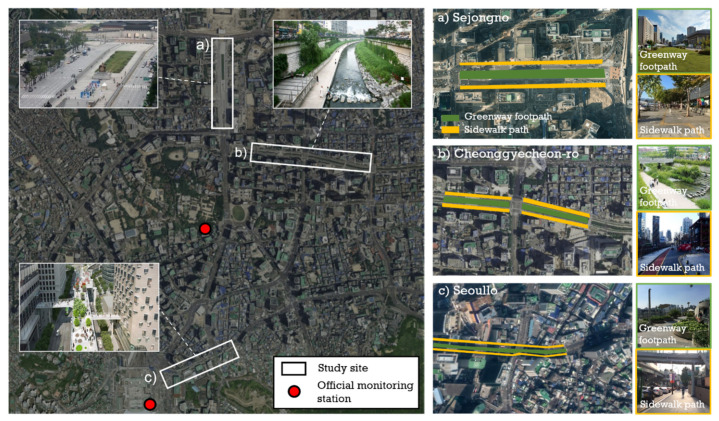
Map of study area.

**Figure 2 ijerph-18-10225-f002:**
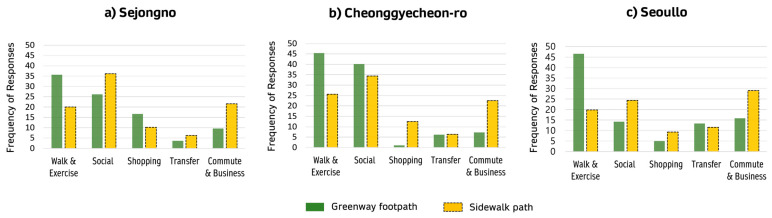
Reasons for visiting the sites.

**Figure 3 ijerph-18-10225-f003:**
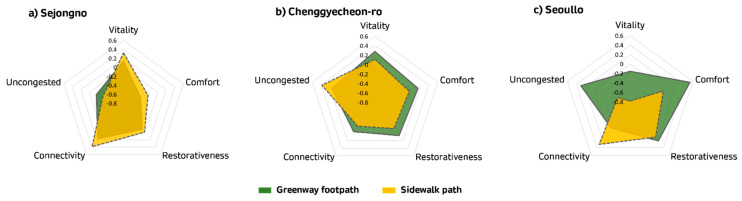
Radar chart comparing street perceptions.

**Table 1 ijerph-18-10225-t001:** Descriptions of the sites.

Attribute	(a) Sejongno	(b) Cheonggyecheon-ro	(c) Seoullo
Regeneration type	Traffic road—pedestrian plaza	Removal of expressway	Overpass—pedestrian path
Height of greenway footpath	Ground level (0 m)	Underground level (−3 m)	Highway level (10 m)
Year opened	2009	2005	2017
Average visitors per year	6,000,000	20,000,000	7,000,000
Number of lanes/ traffic volume	10 lanes/ high	4 lanes/ low	3–12 lanes/ high

**Table 2 ijerph-18-10225-t002:** Demographic characteristics of the sample.

	Full Sample	Greenway Footpath	Sidewalk
Location	621	285 (45.9%)	336 (54.1%)
Sejongno	210 (33.8%)	80 (38.1%)	130 (61.9%)
Cheonggyecheon-ro	220 (35.4%)	94 (42.7%)	126 (57.3%)
Seoullo	191 (30.8%)	111 (58.1%)	80 (41.8%)
Period			
Weekday	332 (53.5%)	146 (51.2%)	186 (55.4%)
Weekend	289 (46.5%)	139 (48.8%)	150 (44.6%)
Gender			
Male	320 (51.5%)	131 (46.0%)	189 (56.3%)
Female	301 (48.5%)	154 (54.0%)	147 (43.8%)
Age			
20s	246 (39.6%)	100 (35.1%)	146 (43.5%)
30s	98 (15.8%)	47 (16.5%)	51 (15.2%)
40s	88 (14.2%)	37 (13.0%)	51 (15.2%)
50s	108 (17.4%)	54 (18.9%)	54 (16.1%)
≥60s	81 (13%)	47 (16.5%)	34 (10.1%)
Frequency of visits			
Almost daily	108 (17.4%)	30 (10.5%)	78 (23.2%)
2–3 times a week	82 (13.2%)	34 (11.9%)	48 (14.3%)
Once a week	73 (11.8%)	35 (12.3%)	38 (11.3%)
Rarely	297 (47.8%)	143 (50.2%)	154 (45.8%)
Never	61 (9.8%)	43 (15.1%)	18 (5.4%)
Purpose of visits			
Commuting	87 (14%)	23 (8.4%)	63 (18.8%)
Shopping	57 (9.2%)	19 (6.7%)	38 (11.3%)
Business	27 (4.3%)	9 (3.2%)	18 (5.4%)
Social activities	188 (30.3%)	76 (26.7%)	112 (33.3%)
To walk or exercise	180 (29%)	121 (42.4%)	59 (17.6%)
Transfer or other	82 (13.2%)	36 (12.7%)	46 (13.7%)
Total	621	285 (45.9%)	336 (54.1%)

**Table 3 ijerph-18-10225-t003:** Constructs and measurement items.

Variable	Description	Variable Type
Greenway footpath	Location of respondent (0 = sidewalk, 1 = Greenway footpath)	Dichotomous
Official monitoring	Observation value from nearest monitoring stations	Continuous
Site	Whether respondent is on the Cheonggyecheon-ro, Seoullo, or Sejongno	Nominal
Weekday	Whether survey was conducted on weekday (0 = No, 1 = Yes)	Dichotomous
Gender	Respondent’s gender (0 = Male, 1 = Female)	Dichotomous
Age	Age of the respondent (1 = 20s, 2 = 30s, 3 = 40s, 4 = 50s, 5 = 60s and older)	Nominal
Visit purpose	Purpose of visit (1 = Commute, 2 = Shopping, 3 = Using business facilities, 4 = Social activities, 5 = For a walk, 6 = Exercise, 7 = Transfer)	Nominal
Walking time	What duration do you walk for on an average on a weekday? (1 = 0–10 min, 2 = 10–30 min, 3 = 30 min–2 h, 4 = more than 2 h)	Ordinal
Asthma	Do you have a respiratory disease (e.g., asthma)? (0 = No, 1 = Yes)	Dichotomous
Smoking	Do you smoke? (0 = No, 1 = Yes)	Dichotomous
Attitude toward the policy	What do you think about controlling traffic to reduce car use and instead promoting pedestrians/walking for a better air quality in downtown Seoul? (1 = Strongly disagree, 6 = Strongly agree)	Ordinal
Sensitivity to air quality	Do you think air pollution significantly affects your health?	Ordinal
Vitality	Value of factor 1	Continuous
Comfort	Value of factor 2	Continuous
Restorativeness	Value of factor 3	Continuous
Connectivity	Value of factor 4	Continuous
Lack of congestion	Value of factor 5	Continuous
Perceived air quality	How do you rate the air quality of this street? (1 = Very bad, 6 = Very good)	Ordinal
Perceived noise level	How do you rate the noise level of this street? (1 = Very quiet, 6 = Very noisy)	Ordinal
Street satisfaction	How satisfied are you with the overall environment of the street? (0 = Not satisfied, 1 = Satisfied)	Dichotomous

**Table 4 ijerph-18-10225-t004:** Air quality perception of the sidewalk and greenway footpath for pedestrians (Mann–Whitney test).

		Air Quality Perception				Street Satisfaction			
Site	*N*	Mean Rank	Sum of Rank	Z-Value	*p*-Value	Mean Rank	Sum of Rank	Z-Value	*p*-Value
Sejongno									
Greenway footpath	84	105.23	8839.00	−0.446	0.655	103.84	8722.50	−0.725	0.468
Sidewalk	130	108.97	14,166.00			109.87	14,282.50		
Cheonggyecheon-ro									
Greenway footpath	97	122.71	11,902.50	−2.012	0.044 **	112.95	10,956.50	−0.201	0.841
Sidewalk	128	105.64	13,522.50			111.27	14,019.50		
Seoullo									
Greenway footpath	121	112.98	13,671.00	−2.461	0.014 **	117.60	14,112.00	−3.965	0.000 ***
Sidewalk	87	92.70	8065.00			85.24	7416.00		

** Significance at 95%, *** Significance at 99%.

**Table 5 ijerph-18-10225-t005:** The factor analysis of the perceived environment factors.

**Factors**	**Formulation of Items**	**Loadings**	**Eigenvalue**	**Explained** **Variance (%)**	**Cumulative** **Variance (%)**	**Cronbach’s** **Alpha**
Factor 1: Vitality	Lively atmosphere	0.744	3.265	15.550	15.550	0.795
	Many cultural elements and attractions	0.740				
	A symbolic place	0.709				
	Interesting and harmonious buildings	0.693				
	Attractive landscape	0.565				
Factor 2: Comfort	Calm and quiet	0.704	3.188	15.181	30.730	0.835
	Clean	0.701				
	Safe from traffic	0.687				
	Convenient and easy to walk on	0.649				
	Comfort	0.621				
	Sufficient rest spots	0.523				
	Well landscaped	0.509				
Factor 3: Restorative-ness	Sufficient trees and shade	0.779	2.535	12.071	42.801	0.757
	Sufficient seating such as a bench	0.771				
	Sufficient green spaces	0.508				
	Not monotonous/boring	0.485				
Factor 4: Connectivity	Sidewalks that are wide enough	0.781	2.043	9.730	52.531	0.705
	Streets that physically connect	0.674				
	No parked cars and bicycles on the streets	0.643				
Factor 5: Lack of congestion	Not noisy	0.815	1.993	9.491	62.022	0.709
	Not congested with traffic	0.785				

**Table 6 ijerph-18-10225-t006:** Results of binary logistic model of street satisfaction.

	Model 1		Model 2		Model 3		Model 4	
	B	Exp (B)	B	Exp (B)	B	Exp (B)	B	Exp (B)
Locational factors								
Greenway footpath	0.309 *	1.363	0.372 *	1.450	0.181	1.198	0.149	1.161
Cheonggyecheon-ro	0.560 **	1.751	0.395 *	1.485	0.098	1.103	0.021	1.022
Seoullo	0.140	1.150	0.022	1.022	−0.156	0.856	−0.153	0.858
Sejongno (Ref)								
Official monitoring	−0.010	0.990	−0.005	0.995	0.003	1.003	0.012	1.012
Weekday	0.199	1.221	0.236	1.266	0.281	1.325	0.281	1.324
Individual factors								
Gender			0.075	0.710	−0.040	0.961	−0.071	0.931
Age in the 20s			0.468 **	1.596	0.396	1.485	0.297	1.346
Age in the 60s or above			−0.113	0.893	−0.108	0.898	−0.134	0.874
Age between 30–50s (Ref)								
Visit for leisure			0.381 *	1.464	0.117	1.124	0.119	1.127
Walking 0–10 min			1.449 **	4.261	1.547 **	4.696	1.467 **	4.338
Walking 10–30 min			0.676 *	1.966	0.774 *	2.168	0.837 *	2.309
Walking 30 min–2 h			0.481	1.618	0.319	1.375	0.241	1.272
Walking more than 2 h (Ref)								
Asthma			−0.543	0.581	−0.433	0.649	−0.362	0.696
Smoking			−0.489 *	0.613	−0.426	0.653	−0.437	0.646
Attitude toward the policy			1.349 ***	3.852	1.693 ***	5.437	1.624 ***	5.074
Sensitivity to air quality			0.315	1.370	0.267	1.306	0.238	1.269
Perceived street environment								
Vitality					0.670 ***	1.953	0.613 ***	1.846
Comfort					0.676 ***	1.967	0.505 ***	1.656
Restorativeness					0.310 **	0.733	0.239 **	0.788
Connectivity					0.128	0.879	0.150	0.860
Lack of congestion					0.426 ***	0.653	0.335 **	0.716
Perceived air quality and noise								
Perceived air quality							0.932 ***	2.539
Perceived noise level							−0.248 **	0.780
Chi-squared (Sig)	15.619 (0.008)		88.628 (0.000)		171.161 (0.000)		200.458 (0.000)	
Nagelkerke	0.033		0.185		0.355		0.408	

* Significance at 90%, ** Significance at 95%, *** Significance at 99%.

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
