# Peer review of "Pedestrianization Impacts on Air Quality Perceptions and Environment Satisfaction: The Case of Regenerated Streets in Downtown Seoul"

_ijerph, 2021, doi:10.3390/ijerph181910225_

Round 1

Reviewer 1 Report

The manuscript "Pedestrianization Impacts on Air Quality Perceptions and Environment Satisfaction: The Case of Regenerated Streets in Downtown Seoul" describes the effect of pedestrianization on individual air quality perception and satisfaction by focusing on three regenerated streets in Seoul, Korea. Although the study does not accept a general application due to restrictions on the number of samples, it has merit regarding the qualitative aspect. 

Details of comments, suggestions, and corrections are attached in the .pdf version of the document.  

Author Response

We thank the reviewer who took the time to provide excellent feedback on our paper and have implemented several key suggestions that have significantly improved the manuscript. We substantially revised the manuscript by rewriting the paragraph and rephrasing the sentences to be clearer. We also used an English-language proofreader for this revision to improve readability of the paper. Specifically, we have indicated what changes have been made to address the comments, having copied and pasted the actual changed text, where relevant. All recommendations made have been incorporated and, in our view, the manuscript is now better able to present the results of our research.

Reviewer 2 Report

The paper is very good in its significance in the field of study.

Nevertheless, the introduction fails to address some of the core themes of the text, i.e. urban regeneration processes impacts in the built environment along with a reflection on the issues of COVID-19 in terms of rate transmissions and its connection with factors such as air quality and the state of (unhealthy) built environment.

So I suggest to add some lines in the introduction, or even add an entire paragraph to address these issues. 

On the urban regeneration processes impacts in terms of healthy city and the "dark side" of urban regeneration, you can base your reflection on the following references but also add more reference on the specific case of Seoul.

Urban regeneration is seen as a sort of “weapon” which enhances social and economic polarization in the wake of rising rents and housing costs, high-standard new developments, and, in general, privatization within the neighborhood

- (2017), Culture-led neighbourhood transformations beyond the revitalisation/gentrification dichotomy. Urban Stud. 54 (4), 953–970.

- (2019). Capital City. Gentrification and the real estate state. London-New York: Verso

- (2019).  From “Ribera Plan” to “Diagonal Mar”, passing through 1992 “Vila Olímpica”. How urban renewal took place as urban regeneration in Poblenou district (Barcelona). Land Use Policy, 89, 10422

On the post-Covid-19 city I suggest to base your reflection on the following references: 

(2020). COVID-19 and Cities: from Urban Health strategies to the pandemic challenge. A Decalogue of Public Health opportunities. Acta Bio Medica Atenei Parmensis, 91(2), 13-22

(2020). The Effects of Air Pollution on COVID-19 Related Mortality in Northern Italy. Environmental and Resource Economics, 76, 611-634

 (2020). Do environmental factors such as weather conditions and air pollution influence COVID-19 outbreaks?. Luxembourg: Publications Office of the European Union.

(2020). New Healthy Settlements Responding to Pandemic Outbreaks: Approaches from (and for) the Global City. The Plan Journal, 5(2): 385-406

OECD (2020). Policy Responses to Coronavirus (COVID-19). Cities policy responses,

(2020) Recommendations for Keeping Parks and Green Space Accessible for Mental and Physical Health During COVID-19 and Other Pandemics. Preventing Chronic Disease, 17, 200204

I look forward to take a glance into the new version of this paper to verify how it improves. 

Author Response

We thank the reviewer who took the time to provide excellent feedback on our paper and have implemented key suggestions that have significantly improved the manuscript. We substantially revised the manuscript by rewriting the paragraph and rephrasing the sentences to be clearer. Specifically, we have indicated what changes have been made to address the comments, having copied and pasted the actual changed text, where relevant. All recommendations made have been incorporated and, in our view, the manuscript is now better able to present the results of our research.

Please see the attchment.

Round 2

Reviewer 2 Report

Thanks for your great work. The paper is ready to be published.